# The Barley Heavy Metal Associated Isoprenylated Plant Protein HvFP1 Is Involved in a Crosstalk between the Leaf Development and Abscisic Acid-Related Drought Stress Responses

**DOI:** 10.3390/plants11212851

**Published:** 2022-10-26

**Authors:** Athina Parasyri, Olaf Barth, Wiebke Zschiesche, Klaus Humbeck

**Affiliations:** Institute of Biology, Martin Luther University Halle-Wittenberg, Weinbergweg 10, 06120 Halle, Germany

**Keywords:** HIPPs, drought, senescence, ABA, crosstalk

## Abstract

The heavy metal associated isoprenylated plant proteins (HIPPs) are characterized by at least one heavy metal associated (HMA) domain and a C-terminal isoprenylation motif. *Hordeum vulgare* farnesylated protein 1 (HvFP1), a barley HIPP, is upregulated during drought stress, in response to abscisic acid (ABA) and during leaf senescence. To investigate the role of HvFP1, two independent gain-of-function lines were generated. In a physiological level, the overexpression of *HvFP1* results in the delay of normal leaf senescence, but not in the delay of rapid, drought-induced leaf senescence. In addition, the overexpression of *HvFP1* suppresses the induction of the ABA-related genes during drought and senescence, e.g., *HvNCED*, *HvS40*, *HvDhn1*. Even though *HvFP1* is induced during drought, senescence and the ABA treatment, its overexpression suppresses the ABA regulated genes. This indicates that HvFP1 is acting in a negative feedback loop connected to the ABA signaling. The genome-wide transcriptomic analysis via RNA sequencing revealed that the gain-of-function of HvFP1 positively alters the expression of the genes related to leaf development, photomorphogenesis, photosynthesis and chlorophyll biosynthesis. Interestingly, many of those genes encode proteins with zinc binding domains, implying that HvFP1 may act as zinc supplier via its HMA domain. The results show that HvFP1 is involved in a crosstalk between stress responses and growth control pathways.

## 1. Introduction

Plants’ sessile lifestyle exposes them to a hostile and ever-changing environment. As a result, they have developed strategies aiming, on one hand, to ensure their growth and reproduction, and on the other hand, to defend themselves against stressful conditions. Environmental stress may be caused by biotic and abiotic factors. Biotic factors originate from living organisms, such as bacteria, fungi, nematodes, insects and other animals. Abiotic stress derives from physicochemical factors, such as drought, flooding, low or high temperature, high light intensity or light deprivation. The response to an upcoming stress involves complex pathways, starting with the perception of the environmental stimuli, which are transferred through a signal transduction network including phytohormones, changes in the ion influx and antioxidant systems, recruitment of specific proteins and signaling molecules, the biosynthesis of secondary metabolites and, on the basis of this, the reprogramming of the gene expression [1]. Since plants are normally exposed to multiple stress conditions at the same time, the different stress response pathways have to be coordinated with each other and harmonized with the developmental programs by molecular factors acting as hubs in crosstalk between the different signaling pathways [2,3]. Phytohormones, such as abscisic acid (ABA), play a major role in this crosstalk between the stress response and the developmental pathways [4,5]. In the present work, it is proposed that *Hordeum vulgare* farnesylated protein 1 (HvFP1) is involved in balancing such trade-offs between the stress responses and the developmental processes.

HvFP1 belongs to the plant specific heavy metal associated isoprenylated plant protein (HIPP) family. Members of this family have a sequence of one or more heavy metal associated (HMA) domain(s), with a core of two cysteines (I/L/MXCXXC) for heavy metal binding, and one C-terminal isoprenylation site (CaaX; [6]). Isoprenylation is a posttranslational modification, which gives a hydrophobic anchor for the protein–protein or protein–membrane interactions in signaling cascades [7]. Some members, including HvFP1 [8], have one or more nuclear localization signals, as well. The exact mode of action of HIPPs is not known, but some studies on *Arabidopsis thaliana* support a function in several stress responses and developmental pathways. Briefly, *AtHIPP26*, which is homologous to barley *HvFP1*, was induced under cold, drought and salt stress, but was downregulated by the ABA treatment and during the developmental leaf senescence [6]. Furthermore, it interacted with the zinc finger homeodomain transcription factor ATHB29, which downstream regulates the drought responsive genes [6,9]. Moreover, AtHIPP3 is a zinc binding protein, which suppressed the flowering in *A. thaliana*, but it was also induced under biotic stress [10].

In contrast to the model plant *Arabidopsis thaliana*, much less is known about HIPPs in other species, especially crop plants. The present investigation aims to unravel the function of the *Hordeum vulgare* HIPP HvFP1. It was first discovered as a cold responsive gene [8]. In the same work, it was found that it is a nuclear localized protein, which may be involved in drought stress, leaf senescence and ABA signaling pathways [8]. To further investigate the role of *HvFP1* in the barley development and abiotic stress responses, two *HvFP1* overexpression (OE) lines were established and studied during optimal (control) and drought stress conditions. Interestingly, the *HvFP1* OE lines exhibited a delay in the developmental leaf senescence. In a transcriptomic analysis with RNA sequencing, possible target genes downstream of HvFP1, were unraveled. In both transgenic lines, the OE of *HvFP1* suppressed the genes involved in the ABA-related drought responses and leaf senescence and positively affected the genes involved in photomorphogenesis and the photosynthetic performance. Some of the major regulators of these pathways, which were affected by the OE of *HvFP1* are zinc binding proteins, e.g., many Far1-related sequences (FRSs), which function as central regulators in photomorphogenesis [11], zinc finger CCCH transcription factors, which have multiple functions in stress responses and development [12], as well as the non-zinc binding ACBD4, which may be involved in ethylene signaling [13].

## 2. Results

### 2.1. HvFP1 Was Expressed during the Developmental Leaf Senescence and in Response to Drought Stress and ABA Treatment

Expression of *HvFP1* was analysed in the WT barley primary leaves, during leaf development and in response to drought and an ABA treatment and its relative transcript level is presented in Figure 1. Plants were grown in soil under controlled conditions. The primary leaf was fully grown on the 13th day after sowing (DAS) and reached its maximum chlorophyll content and photosynthetic activity around the 19th to 21st DAS (Figure 2A). The beginning and the progress of the developmental leaf senescence was illustrated by a decrease in the chlorophyll content and the photosynthetic performance. As shown in Figure 1A, after the onset of senescence, the relative expression level of *HvFP1* increased and was about five times higher than in middle stages of senescence, when more than 50% of the chlorophyll was already degraded. When barley plants were exposed to drought, by withholding irrigation after the 11th DAS, the drought stress caused an earlier and more abrupt induction of *HvFP1* (Figure 1B). Specifically, an up to ten times upregulation was detected after about two weeks of the gradual drying of the soil. The relative expression level was proportional to the progress of the stress, until the drought-induced senescence of the primary leaves after the 27th DAS. The expression level of *HvFP1* remained stable in the control samples (Figure 1B).

Since drought stress responses and leaf senescence are under the regulation of phytohormones, the effect of ABA, salicylic acid (SA), methyl-jasmonate (MeJA), and three cytokinins (kinetin, zeatin and 6-BAP), on the expression of *HvFP1,* was studied at 4 h and 24 h after application (Figure 1C,D). The most prominent effect was observed after the application of ABA, which is the major regulator of drought stress responses. Here, already 4 h after application, the relative transcript level of *HvFP1* was increased by about fourteen times. A much smaller, but still significant increase in the *HvFP1* transcript level was observed after the application of SA and MeJA, while the application of cytokinins, which regulate leaf senescence, suppressed the expression of *HvFP1* after 24 h.

### 2.2. Overexpression of HvFP1 Affected the Time Course of the Developmental Leaf Senescence, but Not of the Drought-Induced Premature Senescence

Expression analyses revealed that *HvFP1* was induced during the developmental leaf senescence and in response to drought. To investigate its function in these processes, two independent, homozygous, OE barley lines, named 21.3I and 21.2A, expressing *HvFP1* 150–200 times higher than the WT, have been generated (transcript and protein levels are shown in Appendix A). The OE construct contains the whole *HvFP1* gene with its two exons and one intron, with one N-terminal strep tag, under the regulation of the double enhanced promoter CaMV35S. The OE of *HvFP1* did not affect the phenotype of the barley plants during their growth (Appendix A). In the experimental design, the time course of both the developmental and the drought-induced senescence was followed by measuring the chlorophyll content and the maximum photosystem II (PSII) efficiency, two sensitive markers of the senescence-specific degradation of the chloroplasts. As shown in Figure 2A, the degradation of the chlorophyll content in the WT primary leaves started on the 27th DAS, and then it was continuously catabolized until the end of the developmental leaf senescence on the 43rd–55th DAS. The PSII efficiency started to decrease at the later stages of the leaf senescence, but still, the PSII centers were rapidly degraded in the WT plants by the 43rd–55th DAS. In contrast to the WT, the onset of the developmental leaf senescence was clearly delayed in both OE lines, as documented by a time lag of four to six days in both, the degradation of chlorophyll and the decrease in the PSII efficiency (Figure 2A). Photos of the primary leaves of the WT and both OE lines, from one approach, are presented in Figure 2B, where the delay in the developmental leaf senescence is obvious for both OE lines.

Prolonged drought stress results in the premature leaf senescence. In our experimental setup, drought stress was induced by stopping irrigation after the 11th DAS and the first time point of drought stress was considered on the 13th DAS. From that time point, the relative water content (RWC) of the soil, which was about 65% in the well-watered controls, continuously decreased, as shown in Figure 2C, reaching 30% after about 10 days and 10% after about 20 days. Under these conditions, which reflected the natural drought stress caused by aridity, a premature senescence was induced in the primary leaves. This was indicated by an early loss of the chlorophyll content, starting on the 21st DAS, and a rapid decrease in the photosynthetic performance, as indicated by a lower PSII efficiency, starting on the 25th DAS (Figure 2C). In contrast to normal senescence, the drastic drought-induced leaf senescence followed the same trend in the WT and both OE lines.

### 2.3. Overexpression of HvFP1 Delayed the Expression of the Senescence Associated Genes, but Also of the Drought Stress-Related Genes

The OE of *HvFP1* caused a delay in the developmental leaf senescence. To test whether the regulation of the senescence related genes was also affected, the expression of the three known senescence associated genes in barley was analysed at various stages of leaf senescence, when the chlorophyll content of the WT primary leaves was at the maximum level (100%) and then reduced to 95, 90, 80, 75 or 50%. Then, the samples from the same time points were used for the expression analysis in both OE lines. All three genes, *HvS40*, *HvSAG39* and *HvSBT*, showed only a basal expression in the mature primary leaves (Appendix A). In the WT, these genes were already significantly induced at the early stages of leaf senescence, when the chlorophyll content was reduced to 90% and their transcript level was considerably increased to 35–60 fold, in the later stages of senescence (Appendix A). The expression level of these senescence associated genes, in both OE lines, was significantly reduced, when compared with the WT samples at each single stage of leaf senescence (Figure 3A).

Despite the fact that there was no difference in the time course of the drought-induced senescence between the WT and OE lines, the expression of the three ABA- and drought- related genes *HvS40*, *HvNCED* and *HvDhn1* was analysed before and during the drought treatment. In the WT, these genes were already induced 8–11 days after the last irrigation of the plants and their expression was drastically increased during the ongoing drought (Appendix A). Interestingly, their expression was clearly delayed in both OE lines, indicating that HvFP1 is functioning upstream of the expression regulation of these drought-related genes (Figure 3B). In addition, we analysed the expression of the ABA-independent *HvHsp17*, which encodes for a small heat shock protein. This gene was also upregulated in response to the drought (Appendix A). However, in contrast to the three drought stress related genes *HvS40*, *HvNCED* and *HvDhn1*, the expression of this gene was not suppressed but rather enhanced in the *HvFP1* OE lines. This significant difference was clearly shown when the gene expression was calculated in the drought samples of both OE lines, in comparison to that of the WT drought samples at each time point (Figure 3B).

### 2.4. Comparative Transcriptomic Analyses of the WT and HvFP1 OE Barley Primary Leaves

As shown before, the OE of *HvFP1* affected the time course of the leaf senescence and the expression of senescence and the drought related genes. This indicates a function of HvFP1 upstream of the regulatory pathways related to stress and senescence. To identify the genes downstream of the HvFP1 function, a comparative transcriptomic analysis of the WT and the OE line 21.3I was performed, under the control and senescence condition, via RNA Seq, with *H. vulgare* cv. Morex version 2 [15] as the reference genome.

#### 2.4.1. Differentially Expressed Gene Analysis

The RNA samples of the WT and the 21.3I lines of the mature (control) and the senescing primary leaves, from the three independent biological replicates, were analysed. The differentially expressed genes (DEG) in the barley primary leaves of the WT and 21.3I lines were estimated by calculating the FPKM (expected number of fragments per kilobase of the transcript sequence per million base pairs sequenced) values. Among the possible sample comparisons, those between the WT and OE control (oe_C vs. wt_C) and the WT and OE senescence (oe_S vs. wt_S) samples, were presented here. The Venn diagrams in Figure 4 showed the total number of up- and downregulated genes, as derived from each comparison, as well as the overlap of the DEGs in both comparisons. The complete list of all genes with their ID, description and log_2_FoldChange is provided in Appendix A. In total, 70 and 105 DEGs, were differentially expressed in oe_C vs. wt_C and oe_S vs. wt_S, respectively. Specifically, 68 genes were upregulated in the OE controls when compared with the WT controls and 93 genes were upregulated in the OE senescing primary leaves compared with the WT senescing primary leaves (Figure 4A). Of these, 50 genes were found specifically upregulated in the OE line 21.3I in both developmental stages. To the contrary, only two genes were downregulated in the OE controls and 12 genes were downregulated in the OE senescing primary leaves, compared with the respective WT samples and of these, only two were found in both developmental stages (Figure 4B).

#### 2.4.2. Functional Enrichment Analysis

The gene IDs and corresponding annotations were obtained from the reference genome *H. vulgare* cv. Morex version 2 [15]. There are still restrictions in the annotation and the GO enrichment analysis for *H. vulgare*. Thus, an additional manual assignment of the gene function was performed, based on the homology of *A. thaliana*, the online databases and the literature. The stringent RNA Seq analysis resulted in the identification of a total of 111 genes, which were upregulated in the *HvFP1* OE line, regardless of the developmental stage of the primary leaves. These genes could be assigned to 16 groups, as presented in Figure 5A. Interestingly, most of these upregulated genes have positive functions in growth and development, as transcription factors/activators, in signaling cascades, in the cell cycle (replication, transcription, translation), in the establishment of the photosynthetic performance and chlorophyll biosynthesis, in the cytoskeleton organization and in the central metabolic processes as polysaccharide, nucleic acid, protein and lipid metabolism. Some of these genes encode zinc-containing proteins and will be discussed later in detail.

Only 12 genes were specifically downregulated in the 21.3I primary leaves, compared to the WT, two of these under control and senescence conditions, and the ten others, only under the senescence condition. Most of these genes have unique functions. As shown in Figure 5B, four genes were assigned for the functions in the cell cycle (replication, transcription, translation), two for transportation and two as senescence associated genes. The single genes were assigned for functions in the signal transduction, phytohormone regulation, stress response and one has an unknown function.

#### 2.4.3. Validation of the RNA Seq Results

The RNA Seq analysis was performed with the samples of the control and the senescing primary leaves of the WT and 21.3I lines. The expression of the specific genes of particular interest was estimated via qRT-PCR, again for the 21.3I line, as well as for the second *HvFP1* OE line 21.2A in the control and the senescing primary leaves (Figure 6). Those genes correspond to the zinc finger CCCH domain containing protein 12 (*HvC3H12*), the acyl-coA binding domain containing protein 4 (*HvACBD4*), one leucine-rich repeat receptor-like kinase (*HvLRR*-*RLK*), one FAR1-related sequence 5 (*HvFRS5*), the auxin response factor 10 (*HvARF10*), the phytochromobilin:ferredoxin oxidoreductase, which is a PFB synthase (*HvPFBS*) and one myb/SANT-like DNA-binding domain protein (*HvMSANTD*). All tested genes showed a significant upregulation in both *HvFP1* OE lines, in comparison to the WT control and senescing leaves.

## 3. Discussion

The present work focused on *Hordeum vulgare* farnesylated protein 1 (HvFP1), which belongs to the family of heavy metal associated isoprenylated plant proteins (HIPPs). This protein family was recently discovered in vascular plants and its members are characterized by two domains: one or more heavy metal associated domain(s) (HMA) and one C-terminal isoprenylation motif [6,16]. The presence of the HMA domain(s) implies a function in either the heavy metal detoxification or the maintenance of heavy metal homeostasis [16,17,18,19,20,21,22]. Recent studies found a connection between the HIPPs and the stress responses of plants, establishing a more complex function of this protein family [6,8,10,23,24,25]. The present work investigated the function of the barley HvFP1 by analysing its expression patterns under normal and stress conditions and studying the effects of the gain-of-function of HvFP1 on the plant responses to the developmental and drought-induced leaf senescence on the physiological and transcriptomic levels.

### 3.1. HvFP1 Is Induced during the Developmental and Drought Induced Leaf Senescence

The expression level of *HvFP1* was analysed during the developmental and drought-induced leaf senescence. The effect of drought stress on barley plants was tested by withholding the water supply after the 11th DAS. This experimental setup reflects natural drought stress conditions, with a slow decrease in the RWC of soil during a prolonged dry period. The expression of *HvFP1* in the control and drought samples of the primary leaves was monitored every two days throughout the experiment, showing a clear induction of the *HvFP1* gene expression during the drought treatment (Figure 1B). This indicates a function of *HvFP1* in the complex response of barley plants to drought. A possible regulatory function of other HIPP members during drought stress has already been established in *A. thaliana*. In fact, *AtHIPP26*, which is the closest ortholog of *HvFP1* in *Arabidopsis*, was also induced under drought stress [6]. In that work, Barth et al. [6] used the yeast two hybrid system and found that AtHIPP26 interacts with the zinc finger homeodomain transcription factor ATHB29, which downstream regulates the stress responsive genes [9]. Furthermore, the presence of the HMA domain in AtHIPP26 was necessary for this interaction. The loss-of-function of *AtHIPP26* resulted in the suppression of some drought-regulated genes, which are normally induced by ATHB29.

The function of *HvFP1* is obviously not limited to the abiotic stress responses, but involves particular aspects of plant development. This is supported by the significant induction of *HvFP1* in the age-dependent leaf senescence, particularly at the late stages of the developmental leaf senescence, when the chlorophyll content was severely reduced (Figure 1A). Interestingly, the homologous to *HvFP1* gene *AtHIPP26* showed different expression patterns, as it was downregulated during the developmental leaf senescence [6], implying a diverse role of the members of the HIPP family in *Arabidopsis* and barley.

### 3.2. Regulation of HvFP1 by the Various Phytohormones

The expression data indicated that in barley, *HvFP1* is involved in the pathways related to drought stress responses and leaf senescence. These pathways are upstream regulated via the action of specific phytohormones. The effect of those molecules on the expression of *HvFP1* was also investigated here. For that, the barley primary leaves were incubated with three stress-related phytohormones, ABA, SA and MeJA [26], and with three cytokinin variants, known to suppress the stress-induced and developmental leaf senescence [27,28]. It has already been reported that the abiotic stress phytohormone ABA [29] strongly upregulated *HvFP1* [8] and its wheat homolog *TaHIPP1* [23], but had no effect on the expression of *AtHIPP26* [6]. This again suggests different functions of the HIPPs in monocots and dicots. Here, *HvFP1* was strongly and quickly upregulated by the ABA in barley, while the biotic stress phytohormones SA and MeJA [30,31] had a much smaller impact on the induction of this gene (Figure 1C). Zhang et al. [23] reported that SA and MeJA led to a negative regulation of *TaHIPP1* after 24 h of treatment and discussed the antagonistic function of ABA and SA, especially under biotic stress. In *H. vulgare*, a negative effect of SA and MeJA on the *HvFP1* expression was not observed.

It is worth mentioning that kinetin, zeatin and 6-BEP, which belong to the isoprene (ISCK) or aromatic (ARCK) classes of cytokinins [28], downregulated *HvFP1* after 24 h of treatment (Figure 1D). Cytokinins are involved in many aspects of plant development, including leaf senescence. It has been reported that the exogenous application of cytokinins or the OE of cytokinin biosynthesis genes led to the delay in senescence [28]. Actually, *trans*-zeatin had the highest effect in delaying the wheat leaf senescence, followed by kinetin and 6-BAP [32]. Moreover these phytohormones were endogenously decreased under stress [33], while ABA, SA and MeJA increased in the response to abiotic or biotic stress [5,34,35]. It is well established that the interplay among various phytohormones, such as ABA and cytokinins [27], determines the plant responses under specified conditions. Here, one assumption is that under abiotic stress, the increase of ABA, in combination with the decrement in cytokinins, favors the induction of *HvFP1*.

This complex expression pattern might reflect a more sophisticated role of HvFP1 in different stress- and development-related pathways, similarly to the hubs involved in the crosstalk between different pathways. Plants have developed complex and highly flexible signaling pathways, interconnected for balancing specific processes, such as stress responses with the control of growth and development. One way to study the mode of the function of HvFP1 in such pathways was to establish and study the transgenic lines, which overexpress the gene of interest.

### 3.3. Overexpression of HvFP1 Causes Distinct Changes in the ABA- and Stress-Related Gene Expression and in the Course of Leaf Senescence

The study of the *HvFP1* OE lines 21.3I and 21.2A under the drought-induced and developmental leaf senescence gave interesting results regarding the mode of action of this HIPP. Under drought stress, the expression of the typical ABA-related, drought-induced genes *HvS40*, *HvNCED* and *HvDhn1* was clearly suppressed when compared to the WT (Figure 3B and Appendix A). The expression of those well-established drought regulated genes depends on the severity of the stress [36,37]. Specifically, *HvS40* was regulated by ABA under abiotic stress, including water deprivation [38]. *HvNCED* encodes for a 9-*cis*-epoxycarotenoid dioxygenase, which is a key rate limiting enzyme in the ABA biosynthesis [36]. It was shown that rice plants overexpressing *OsNCED3* exhibited an enhanced drought tolerance [39]. Then, *HvDhn1*, which belongs to the LEA group II family, was induced under drought stress and encodes for the proteins with the protective function against the deleterious effects of dehydration [40]. All genes were upregulated in the drought samples after the 21st DAS, but the induction of these three genes was clearly suppressed in both OE lines 21.3I and 21.2A (Figure 3B and Appendix A).

Interestingly, the expression of the drought-induced gene *HvHsp17* followed a different expression pattern among the barley lines. In the WT drought samples, *HvHsp17* was induced after the 21st DAS and this induction was higher with the progress of the drought stress, but its transcript level in the OE lines was equal or even higher than the WT samples on the same time points (Figure 3B and Appendix A). This opposite trend may be linked to the action of ABA, which positively regulates *HvS40* [38], *HvNCED* [36] and *HvDhn1* [40], but has no effect on *HvHsp17*. More specifically, the study of the Hsp family in rice showed that the ABA treatment could affect the expression of specific heat shock genes, but had no impact on the small heat shock protein Hsp17 [41]. Then, Sun et al. [42] noted even a suppression of *AsHsp17* in *Agrostis stolonifera* leaves after a treatment with ABA and the downregulation of ABA biosynthesis genes in *AsHsp17* OE lines, which implies a negative regulation of ABA on the expression of *AsHsp17*. They proposed an AsHSP17-mediated, ABA-independent stress signaling under abiotic stress through the DREB1/CBF- and DREB2-related transcription factors.

Furthermore, the developmental leaf senescence caused an induction of the ABA-related genes *HvS40*, *HvSAG39* and *HvSBT*, when the chlorophyll content dropped to 90% in the WT primary leaves (Appendix A). It is known that the nuclear localized HvS40 is regulated by HvWHIRLY1 during developmental senescence and its expression was highly influenced by ABA [38,43]. Furthermore, mutants of *AtS40* showed a delayed senescence and a downregulation of the senescence associated genes [44,45]. Then, rice *SAG39* encodes for a cysteine protease and was found to be highly induced in the response to various stimuli, including the ABA treatments, implying a role in the ABA-induced leaf senescence [46]. Another type of proteases, the subtilisin-like serine proteases or subtilases, are induced during leaf senescence for the degradation of proteins [47]. Wang et al. [48] found that one member of the subtilisin family in *Arabidopsis* is responsible for the degradation of OPEN STOMATA 1 (OST1) under drought stress, for the regulation of the ABA signaling. Here, in both *HvFP1* OE lines, the induction of *HvS40*, *HvSAG39* and *HvSBT* was clearly suppressed during the developmental leaf senescence (Figure 3A and Appendix A).

On a physiological level, the effect of the *HvFP1* OE was obvious only during the developmental leaf senescence (Figure 2A,B). In both OE lines, a delay in the reduction of the chlorophyll content and the PSII efficiency of the primary leaves was noted, leading to a late developmental senescence (Figure 2A,B). In fact, the PSII efficiency started to decrease only at later stages, indicating that the efficiency of the remaining PSII centers was still high, during the first phase of the chlorophyll degradation. This is a well-known phenomenon [49,50]. Nevertheless, from that point on, the PSII efficiency declined rapidly until the completion of the senescence of the primary leaves and the inactivation of all PSII centers. This process was faster under the extreme condition of drought stress, in order to ensure the survival of plants, until water becomes available again [51]. As outlined in the review from Munné-Bosch and Alegre [51], phytohormones, especially ABA, are involved in the regulation of this drastic process, which allows the remobilization of the nutrients from the old parts of the plant for the sake of young growing tissue, and in addition, helps to avoid losses through transpiration via the old leaf. One could assume that if *HvFP1* is involved in an ABA-dependent signaling pathway, it could influence the onset of the developmental leaf senescence, since ABA is a positive regulator of this process [4], and through that, is crossing the ABA-dependent signaling pathway for the regulation of the drought stress responses.

### 3.4. Identification of the Target Genes of the HvFP1 Regulatory Pathways

A transcriptomic analysis by RNA Seq gave more information regarding the genetic reprogramming in excess of HvFP1. Due to the obvious phenotype, during the developmental leaf senescence, mature (control) and senescent WT and OE primary leaves were chosen for analysis. A comprehensive analysis of the gene expression in the primary leaves of the OE line with that in the WT, resulted in a total of 123 DEGs, regardless of the control or senescence state (Figure 4). Interestingly, only 12 genes were downregulated, implying a rather positive regulation of the genes downstream of the HvFP1 functional network during the control and senescent conditions. Regarding the downregulated genes, not much information was extracted. One third of them regulate the functions of the cell cycle, especially the DNA transcription and translation. Then, two SAGs were downregulated and it can be assumed that this is related to the delay in the developmental leaf senescence in the HvFP1 OE lines. Two other genes are functioning in transportation, especially one transporter for the cadmium detoxification and one for the intracellular protein transportation. Only one gene was found in each functional group for signal transduction, phytohormone regulation, stress responses and an unknown/other function (Figure 5B).

The list of 111 upregulated genes in the *HvFP1* OE line contained various representatives, which were manually sorted, according to their predicted function (Figure 5A). Interestingly, a number of genes, which were positively regulated in the OE line, correspond to multiple zinc binding FAR1-related sequence (FRS5) transcription activators, zinc finger CCCH domain-containing transcription factors and zinc finger containing genes for posttranslational modifications (Appendix A). The presence of the zinc binding domain(s) in these proteins hints at a possible function of HvFP1 in delivering zinc via its HMA domain for their downstream activation. Then, they could downstream regulate the central plant responses, such as in the case of FRS5s, which are involved in the phyA signaling, photomorphogenesis, chlorophyll biosynthesis, chloroplast development, circadian rhythm, ROS homeostasis, ABA signaling, abiotic stress responses and leaf senescence [11,52]. Specific members of the FRS family are found in the same signaling pathway with the auxin response factor 10 (ARF10) [11] and one phytochromobilin:ferredoxin oxidoreductase (PFBS), which are both strongly induced in the HvFP1 OE lines (Figure 6). The tomato homolog ARF10 is involved in the fruit development and ripening through the chlorophyll accumulation, seed dormancy and germination through the regulation of ABSCISIC ACID INSENSITIVE 3 (ABI3) in the ABA signaling pathway [53]. PFBS is important for the chromophore activation in phytochromes and the downstream regulation of multiple aspects of plant development [11,54,55]. Then, the barley zinc binding CCCH transcription factors are also known to play important regulatory roles in both biotic and abiotic stress responses and in the developmental processes [12]. The hypothesis of the zinc transportation by HvFP1 to these zinc-containing proteins to regulate their function, is further supported by previous studies on *Arabidopsis*, which found that AtHIPP26 interacts with the zinc finger homeodomain transcription factor ATHB29 for the regulation of some stress response genes [6,9] and that the presence of the HMA domain is important for this interaction [6], while the biotic stress induced AtHIPP3 was found, via ICP-MS, to bind zinc in its HMA domains [10]. Furthermore, in a recent study on *Chenopodium quinoa* plants, CqHIPP34, which is homologous to AtHIPP26, was found to interact with the zinc finger homeodomain transcription factor CqZF-HD14 for the regulation of the drought stress responses [56]. However, whether HIPPs, such as HvFP1, act via this mechanism, has to be clarified in future experiments.

It is worth mentioning that the OE of *HvFP1* resulted in a positive regulation of the components for the posttranslational modifications sumoylation and palmitoylation. Sumoylation is an important modification for the abiotic stress signaling and responses [57], probably through the regulation of the ABA signaling pathway [58]. Then, palmitoylation involves the action of a DHHC-type zinc finger protein, which was positively regulated in the OE *HvFP1* line and the study of this gene family, in other plant species, showed that it regulates the shoot branching in *Arabidopsis* [59] and increased the tillers and overall grain yield in rice [60]. HvFP1 seems to act upstream of both protein modifications, which regulate plant stress responses and development.

In addition to the zinc finger containing genes, the *HvACBD4* was also strongly upregulated in both *HvFP1* OE lines (Figure 6). The particular interest in this gene originates from the study of Gao et al. [61], who studied this protein family in *A. thaliana* and found that another member, AtACBP2, interacts with AtFP6 (also known as AtHIPP26). The same team suggested that specific ACBPs, which possess ankyrin repeats (like AtACBP2), or kelch domains (like AtACBP4 and its barley homolog HvACBD4), promote protein–protein interactions in order to provide acyl-CoA esters to enzymes without acyl-CoA binding domains or, interestingly, to transfer heavy metals to the transcription factors through the interaction with the HMA domain-containing proteins, such as AtHIPP26 or, in the present work, HvFP1 [13,61]. Furthermore, AtACBP2 and AtACBP4 are known to have one mutual interaction partner, the ethylene binding protein (EBP) [13,61]. One hypothesis is that the interaction of ACBPs with heavy metal binding proteins could have a regulatory function by affecting the Cu^2+^-mediated ethylene binding and signaling through AtEBP [13,61,62]. Finally, some members of the ACBP family are involved in the abiotic stress responses and leaf senescence [63], through the regulation of specific phytohormones. Specifically, AtACBP2 was induced by ABA and conferred resistance to drought stress and enhanced the ABA-mediated leaf senescence [64].

Taking the above results together, a model is presented in Figure 7, with a summary of the already discussed genes and the proposed mode of action of HvFP1. The present study showed that HvFP1 was induced during drought stress, leaf senescence and in response to ABA. Moreover, the OE of *HvFP1* suppressed the drought-induced genes, *HvNCED*, which encodes the key enzyme of the ABA biosynthesis, *HvS40* and *HvDhn1*, as well as the senescence associated genes *HvSAG39* and *HvSBT*. Such negative feedback loops in the ABA signaling have already been described [65,66], which are important in balancing the major developmental processes and stress responses, and by this improving the fitness of the plants. Furthermore, the OE of *HvFP1* resulted in the delay of the developmental leaf senescence and in the induction of the genes needed for growth, e.g., genes involved in photosynthesis and chlorophyll biosynthesis. Recent findings underline that such reciprocal balancing and fine-tuning of both the growth and abiotic stress responses, is beneficial for plant survival [2,3]. The multifunctional mode of action of HvFP1 is supported in the present work by a study of the OE lines, transcriptome analysis, as well as promoter analysis. A search for the cis-binding regulatory elements upstream of *HvFP1,* revealed typical promoter motifs (i.e., TATA-box and CAAT-box), as well as phytohormone response elements (ABA, MeJA, GA and auxin), transcription factor binding sites (WRKY, MYB and MYC), stress response motifs (cold, dehydration, anoxia and defense) and developmental motifs (light responses, xylem expression, meristem activation, zein metabolism and seed regulation) (Appendix A). Future experiments are necessary in order to confirm the interaction partners of HvFP1 and the role of zinc in its function.

## 4. Materials and Methods

### 4.1. Plant Cultivation

Barley *Hordeum vulgare* cv. Golden Promise WT or transgenic plants were used in all experimental approaches. Barley seeds were spread on wet paper and covered with aluminum foil. They were stratified at 4 °C for 96 h and germinated at 23 °C/18 °C in a 16 h/8 h thermoperiod for 48 h in the dark. The germinated seeds were sown in 5 L Mitscherlich pots containing soil ‘Werkverband typ ED73’, pH 5.8 (Einheitserdewerke Werkverband e.V., Germany) without fertilizers. The plants were grown under controlled, long day conditions in greenhouse cabinets with 16 h light 23 °C/8 h dark 18 °C, with a light intensity of 100 μmol m^−2^ s^−1^ and 45% relative humidity.

### 4.2. Drought Stress

For the drought stress, 10 germinated barley seeds of the WT or transgenic lines were sown in 1.5 kg soil (DE70) in each Mitscherlich pot and irrigated with 0.6 L water to reach a soil RWC of 65%, at the beginning of the experiment. The plants were grown in greenhouse cabinets and the last irrigation of all plants was on the 11th DAS. Then, the drought stress was applied by withholding water, while the control plants were irrigated every two days, by weighting the pots and adding the same amount of missing water in order to maintain the soil RWC at 65%. Every two days, the Mitscherlich pots were rotated around the cabinet, in order to eliminate the effect of the location on the results. At the same time points, the changes in the physiological parameters were monitored by measuring the PSII efficiency and the chlorophyll content of 20 primary leaves. The samples of four to five primary leaves were taken at each time point, they were frozen in liquid nitrogen and stored at −80 °C. The experiment was performed three times.

### 4.3. Developmental Senescence

For monitoring the developmental leaf senescence, 10 germinated barley seeds of the WT or transgenic lines were sown in soil in Mitscherlich pots and were grown in greenhouse cabinets. Every two days, the Mitscherlich pots were rotated around the cabinet, in order to eliminate the effect of the location on the results. At the same time points, the changes in the physiological parameters were monitored by measuring the PSII efficiency and the chlorophyll content of 20 primary leaves, until the senescence of the primary leaves. The samples of four to five primary leaves were taken at each time point, they were frozen in liquid nitrogen and stored at −80 °C. The experiment was performed three times.

### 4.4. Phytohormone Treatments

The germinated barley seeds of the WT plants were sown in Mitscherlich pots and were grown in greenhouse cabinets. On the 13th DAS, the primary leaves were cut and incubated in 50 mL of a phytohormone solution. ABA, SA and MeJA were first dissolved in pure ethanol and then diluted in tap water in a final concentration of 100 μM ABA, 1 mM SA, 200 μM MeJA. Cytokinins were first dissolved in 1 M KOH and then diluted in tap water in a final concentration of 50 μM kinetin, 50 μM zeatin and 50 μM 6-BAP. In the control primary leaves, either tap water with pure ethanol (final concentration 0.05%) or tap water with KOH (final concentration 80 µM) was applied. The primary leaves were harvested at 0, 4 and 24 h after the incubation in the phytohormone solution. The samples of four to five primary leaves were taken at each time point, they were frozen in liquid nitrogen and stored at −80 °C. The experiment was performed three times.

### 4.5. Photosystem II Efficiency

A MINI-PAM fluorometer (Walz GmbH, Effeltrich, Germany) was used to measure the photosynthetic efficiency of the primary leaves in the different treatments. The leaf area was covered with appropriate clips for 5 min for the dark adaption. Then, F_0_ was measured using a weak pulsed measuring light, the MINI-PAM provided a strong saturating light flash to measure F_m_ and the PSII efficiency was calculated as the ratio F_v_/F_m_, where F_m_ was the maximum fluorescence and F_v_ was the variable fluorescence (F_v_ = F_m_ − F_0_).

### 4.6. Chlorophyll Content

A SPAD-502 instrument (Soil Plant Analysis Development, Konica Minolta Sensing Europe B.V., Munich, Germany) was used to measure the chlorophyll content of the primary leaves in the different treatments. The chlorophyll content is expressed in SPAD units.

### 4.7. Isolation of Total RNA

The total RNA was isolated with TRIzol reagent, according to the method described by Chomczynski and Mackey [67]. The concentration of the total RNA was estimated using a Nanospectrophotometer (NanoPhotometer^®^ NP80, Implen, Munich, Germany).

### 4.8. Synthesis of Complementary DNA (cDNA)

The cDNA synthesis was performed with the RevertAid^TM^ H Minus First Strand cDNA Synthesis kit (Thermo Fisher Scientific, Waltham, Massachusetts, USA) with 1 μg RNA, according to the kit instructions, in a Thermocycler Professional Trio, Biometra (Analytik, Jena, Germany).

### 4.9. Quantitative Real-Time PCR (qRT-PCR)

The qRT-PCR was performed with four dilution series for each sample, corresponding to 1:4, 1:16, 1:64 or 1:256, in order to test the efficiency of the primer pairs. In each reaction, 2 μL template cDNA was mixed with 2.2 μL DEPC-treated water, 5 μL SYBR green master mix (KAPA SYBR FAST Universal, KAPABIOSYSTEMS) and 0.4 μL of each gene-specific primer (5 μM). The gene IDs, gene names, primer names and sequences and amplicon sizes are provided in the Appendix A. The qRT-PCR reaction was carried out using a CFX Connect Real-Time PCR Detection System (Bio-Rad, Laboratories GmbH, Hercules, CA, USA). The software of the cycler estimated the CP values and the slopes of regression line for calculating the PCR efficiency. The relative expression level and the standard errors were calculated using the REST-384 © 2006 software (Relative Expression Software Tool-384, version 2; Qiagen GmbH, Hilden, Germany; [14]), normalized to the reference genes *HvPP2A* [68,69], *HvActin* [68,70] and *HvGCN5* (which showed a constant expression under all conditions tested).

### 4.10. Sample Preparation for the RNA Sequencing

The total RNA of the selected samples was extracted with TRIzol reagent, as described above [67]. Then, one additional purification step was performed with the RNeasy^®^ Mini Kit, RNA Cleanup protocol (Qiagen, Hilden, Germany), in order to increase the purity of the RNA. The quantity and quality of the samples was estimated with a Nanospectrophotometer (Implen, Munich, Germany) and an Agilent RNA 6000 Pico Kit (Agilent Technologies, Santa Clara, CA, USA) in a Bioanalyzer2100 system (Agilent Technologies, Santa Clara, CA, USA), which provided the concentration of RNA, the subunits of ribosomal RNA, the ratio of 25S and 18S rRNA and the RNA Integrity Number (RIN) for each sample. Following the quality control, a total 20 µL of each good quality RNA sample with a concentration > 40 ng/µL were sent to Novogene Co., Ltd. (Cambridge, UK) for the library preparation and sequencing, according to their protocols and standards.

### 4.11. Mapping to Reference the Genome and Quantification

For mapping to the reference genome, the version 2 of *H. vulgare* cv. Morex genome annotation [15] was recruited as the reference genome and the HISAT2 software was used for the mapping. The quantification of the total transcripts was accomplished by counting the number of reads mapped to each gene with the featureCounts program. Furthermore, the gene length and sequencing depth were taken into consideration and the FPKM (fragments per kilobase of transcript per million mapped reads) value was estimated. These values reflect the gene expression level.

### 4.12. Differential Expression Analysis

The samples from the three independent biological replicates were used for the RNA Seq analysis. For this reason, the DESeq2R package was used for the calculation of the differential expression level. DESeq2 depends on a model based on the negative binomial distribution, in order to determine the differential expression and the statistical significance. The estimated *p* values were further adjusted with the Benjamini and Hochberg’s approach for the false discovery rate (FDR). Only the genes with log_2_FoldChange > 2 and adjusted *p*-value < 0.05 were included in the lists with the differentially expressed genes. The Venn diagrams were created for the presentation of the DEGs using the online page of InteractiVenn [71].

### 4.13. Functional Analysis

The functional annotation of each gene was analysed for its homologies and biological processes with Uniprot (https://www.uniprot.org/; accessed on 29 October 2021), Barlex (https://apex.ipk-gatersleben.de/apex/f?p=284:10::::::#HOME_LINK#; accessed on 9 September 2021), Arabidopsis Information Resource—TAIR (https://www.arabidopsis.org/; accessed on 14 December 2021) and NCBI blast (https://blast.ncbi.nlm.nih.gov/Blast.cgi; accessed on 30 September 2021).

### 4.14. Promoter Analysis

The plant ensemble database (http://plants.ensembl.org/Hordeum_vulgare/Info/Index; accessed on 26 August 2022) was used to retrieve the sequence of the 2 kb upstream of 5′ untranslated region (UTR) of *HvFP1*. This sequence was introduced to the PlantCARE bioinformatics tool [72] for detecting the *cis*-acting regulatory elements.

### 4.15. Statistical Analysis

Each experimental approach included three or four independent biological replicates and twenty technical replicates. The qRT-PCR analysis was performed with samples from three independent biological experiments, with four analytical replicates for each sample. For the qRT-PCR results, the statistical analysis and the *p* values were determined by using the Pair Wise Fixed Reallocation Randomisation Test ©, included in REST-384 © 2006 (Relative Expression Software Tool—384, version 2.; Qiagen GmbH, Hilden, Germany; [67]). The RNA sequencing analysis was performed with samples from three independent biological replicates and the *p* values of differential gene expression were calculated with the DESeq2R package and adjusted with the Benjamini and Hochberg’s approach for the false discovery rate (FDR).

## 5. Conclusions

The present work showed that the HvFP1 HIPP is regulated by ABA and is involved in the drought-induced and developmental leaf senescence. In fact, the OE of this gene resulted in the physiological delay of the leaf senescence and influenced the expression of the drought- and senescence-related genes. In addition to these results, an RNA Seq analysis supports a function of HvFP1 in hubs connecting various signaling pathways for the reciprocal regulation of the stress responses and growth control. The presence of the HMA domain indicates that HvFP1 could act as a zinc chaperon targeting the zinc-binding regulatory factors.

## Figures and Tables

**Figure 1 plants-11-02851-f001:**
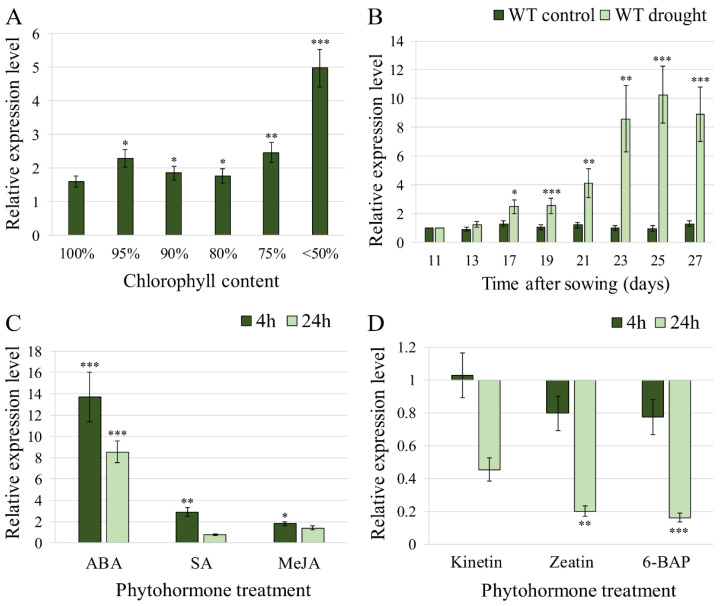
The relative transcript level of *HvFP1* in the samples of the WT barley primary leaves: (**A**) during different stages of the developmental leaf senescence, as defined by the reduction in the chlorophyll content, compared with the samples from the 13th DAS (set as 1); (**B**) at different time points of drought stress, compared with the well-watered samples from the 11th DAS (set as 1); (**C**) after treatment with abscisic acid (ABA), salicylic acid (SA) and methyl-jasmonate (MeJA) in comparison to the control treatment without phytohormones (set as 1) and (**D**) after treatment with kinetin, zeatin and 6-benzylaminopurine (6-BAP), in comparison to the control treatment without phytohormones (set as 1). Mean relative expression level of the three independent biological replicates, standard errors and p-values were determined by REST-384 © 2006 (Relative Expression Software Tool-384, version 2.0, [14], Qiagen GmbH, Hilden, Germany) and normalized against *HvPP2A*, *HvActin* and *HvGCN5*. Statistically significant differences between (**A**) the WT samples at various developmental stages, compared to the WT samples from the13th DAS, (**B**) control and the drought WT samples, compared to the well-watered WT samples on the 11th DAS, and (**C**) WT samples after treatment with various phytohormones, compared to their respective controls, are indicated with asterisks: *p* < 0.05 (*), *p* < 0.01 (**), *p* < 0.001 (***).

**Figure 2 plants-11-02851-f002:**
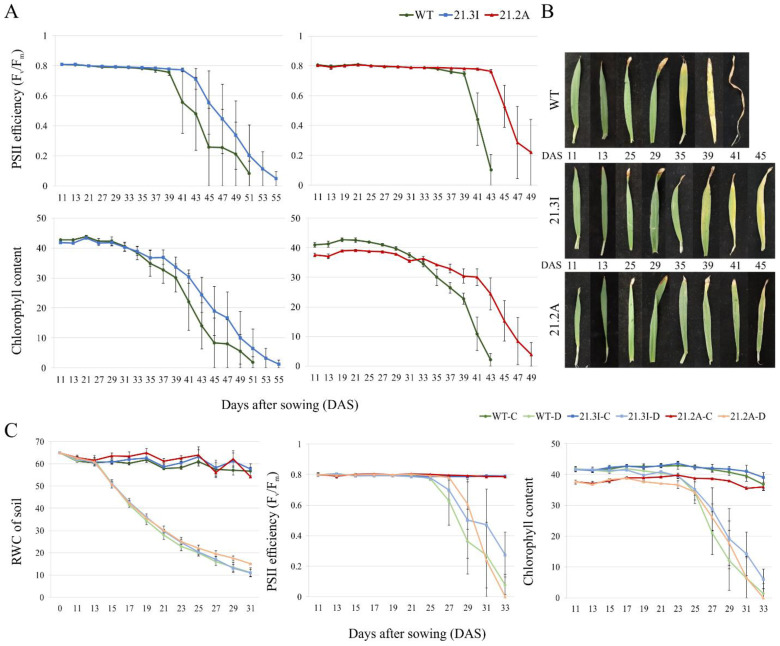
Study of the barley WT and the *HvFP1* OE lines during leaf senescence and under drought stress: (**A**) The maximum photosystem II (PSII) efficiency (F_v_/F_m_ ratio) and the chlorophyll content (in SPAD units) of the WT, 21.3I and 21.2A lines, in the course of leaf senescence; (**B**) Photos of the primary leaves of the WT, 21.3I and 21.2A lines, from one exemplary senescence approach and (**C**) The relative water content (RWC) of the soil during the drought stress approaches, which was maintained at 65 % at the beginning of the experiment, the maximum PSII efficiency (F_v_/F_m_ ratio) and the chlorophyll content (in SPAD units) of the WT, 21.3I and 21.2A primary leaves in the control (C) and drought (D) treatments. The graphs show the mean values and the standard errors of the three independent biological replicates.

**Figure 3 plants-11-02851-f003:**
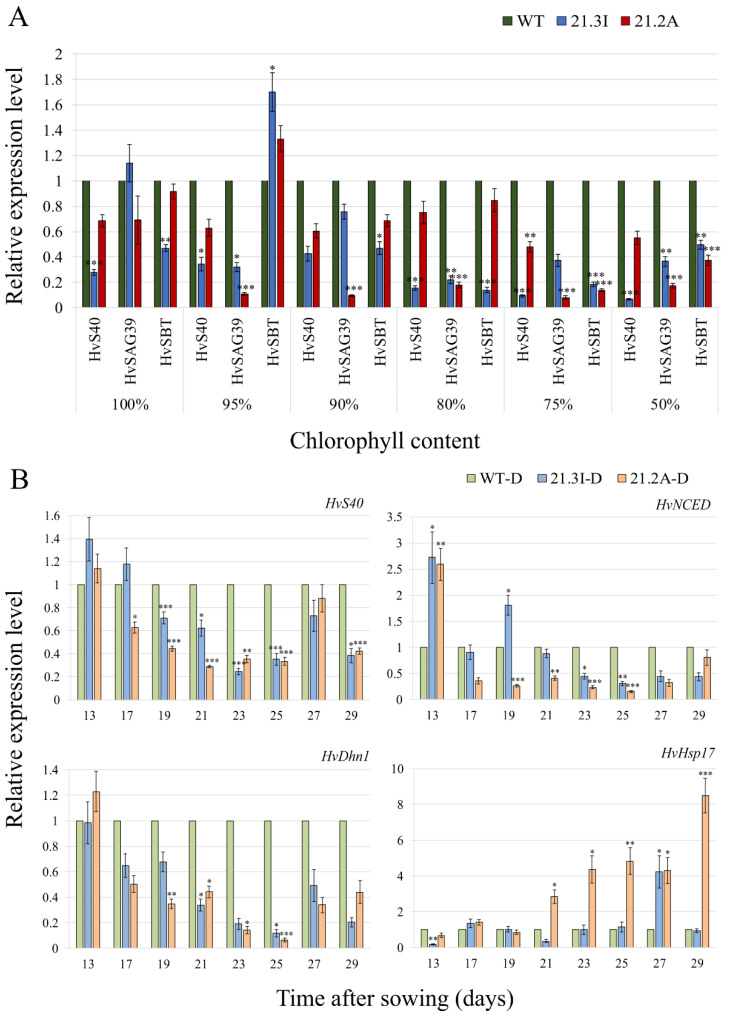
The relative transcript level of: (**A**) senescence associated genes in the two *HvFP1* OE lines, compared with the samples of the WT at each developmental stage (set as 1; stages are defined by the chlorophyll content of the WT primary leaves) and (**B**) drought related genes in the drought (D) samples at different time points of the drought induced senescence, compared with samples of the WT-D at each time point (set as 1). Mean relative expression level of the three independent biological replicates, standard errors and p-values were determined by REST-384 © 2006 (Relative Expression Software Tool-384, version 2.0, [14]),Qiagen GmbH, Hilden, Germany and normalized against *HvPP2A*, *HvActin* and *HvGCN5*. Statistically significant differences between (**A**) the samples of the OE lines, compared to the samples of the WT at each developmental stage and (**B**) the drought samples of the OE lines compared to the drought samples of the WT at each time point, are indicated by asterisks *p* < 0.05 (*), *p* < 0.01 (**), *p* < 0.001 (***).

**Figure 4 plants-11-02851-f004:**
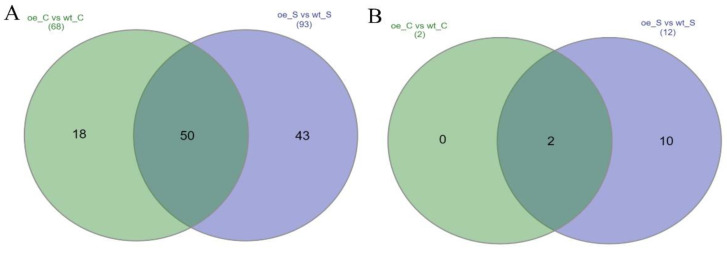
Venn diagrams of: (**A**) upregulated genes in the oe_C vs. wt_C and the oe_S vs. wt_S samples and (**B**) downregulated genes in the oe_C vs. wt_C and the oe_S vs. wt_S samples.

**Figure 5 plants-11-02851-f005:**
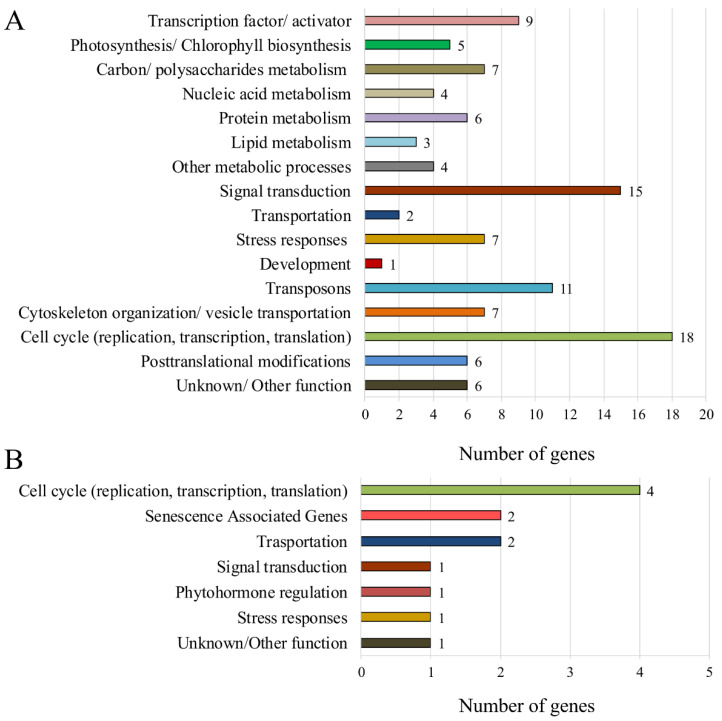
Bar charts with the functional annotation groups and the number of: (**A**) upregulated genes and (**B**) downregulated genes, in the mature and/or senescing primary leaves of the *HvFP1* OE line.

**Figure 6 plants-11-02851-f006:**
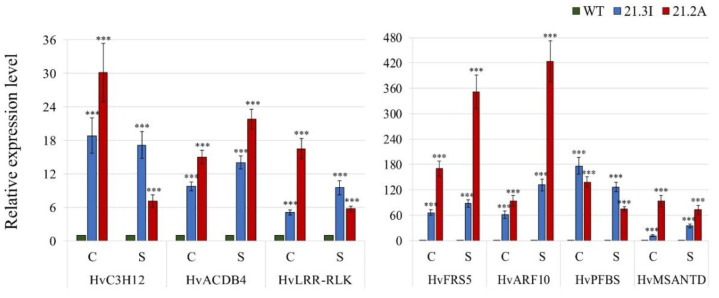
The relative transcript level of the selected genes, as derived from the RNA Seq analysis: *HvC3H12*, *HvACBD4*, *HvLRR*-*RLK*, *HvFRS5*, *HvARF10*, *HvPFBS* and *HvMSANTD*, in the control (C) and senescing (S) samples, compared with the corresponding WT-C or WT-S samples (set as 1). Mean relative expression level of at least three independent biological replicates, standard errors and p-values were determined by REST-384 © 2006 (Relative Expression Software Tool-384, version 2.0, [14], Qiagen GmbH, Hilden, Germany) and normalized against *HvPP2A*, *HvActin* and *HvGCN5*. Statistically significant differences between the control samples of OE lines, compared to the WT control leaves and the senescing samples of the OE lines, compared to the WT senescing leaves are indicated by asterisks, *p* < 0.001 (***).

**Figure 7 plants-11-02851-f007:**
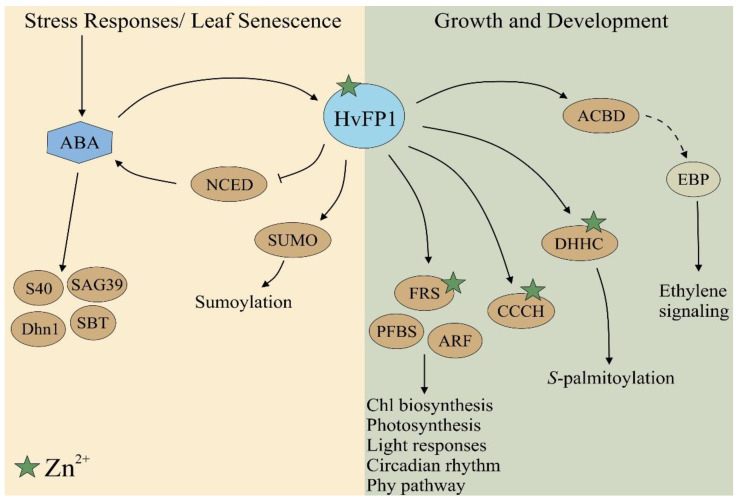
Proposed model for the mode of action of HvFP1 in stress, senescence and growth regulating pathways. Interaction with other proteins (shown as ellipses) and ABA (shown as hexagon) is pictured by arrows (for the positive effect) and bars (for the negative effects). Proteins containing zinc are marked with a green star.

## Data Availability

Not applicable.

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
