# Peer review of "The Barley Heavy Metal Associated Isoprenylated Plant Protein HvFP1 Is Involved in a Crosstalk between the Leaf Development and Abscisic Acid-Related Drought Stress Responses"

_plants, 2022, doi:10.3390/plants11212851_

Round 1

Reviewer 1 Report

The obtained results are interesting and the discussion is convincing. However, presentation and scientific style can be significantly improved to increase perceptibility and overall quality of the paper.

Do not use abbreviations in the title (HIPP, ABA). Rephrase as "...in a crosstalk between..."

Abstract

Contains unexplained abbreviations.

Do not divide compound words with a hyphen in the Abstract and further in the whole text. It is "crosstalk", "upregulated", "overexpression", "nonliving" (although better use "physicochemical"), "feedback", "photomorphogenesis", "posttranslational", "downregulated", "multifunctional", "upregulation", "setup".

However, it is "ever-changing", "well-known", "trade-off", "well-watered", "drought-induced", "fine-tuning" as used correctly.

Introduction

The aim needs to be formulated in more detailed manner, including also model species. "Normal" is not a scientific term, probably "optimal conditions" or "well-watered" are meant.

In the last paragraph, no need to describe obtained results in detail, moving even to a discussion.

Do not introduce abbreviations that are not used further in the text (as NLS).

Results

Do not use "das" as abbreviation for "days after sowing", it is confusing, use "DAS" when necessary.

Use past tense when describing the results (lines 83-84, lines 87–90, line 94, line 101, line 111, line 122, line 124–126, line 129, line 136–143, lines 151–157, lines 163–165, lines 168–173, lines 223–225).

Do not use references to literature (line 133).

Indication of statistically significant differences in Figure 1, Figure 3, Figure 6 seems to be incorrectly used and are confusing, better report results of pairwise post-hoc analysis as different letters, when p < 0.05.

In Figure 2, Figure 3B, legends on X axis should be "Time after sowing (days)". These graphs are too small for optimum perception. 

Section 2.4 contains redundant experimental details belonging to Materials and methods.

In Discussion, use past tense when describing the results obtained in the present study.

Materials and methods

The type of germination conditions can be indicated as "thermoperiod" (lines 479–480).

For a drought stress experiment (and other separate experiments), how many times exactly these experiments were performed (it cannot be "at least three times"). Or does this mean that there were three biological replicates in a single experiment?

Use "." for decimals instead of ",".

Use correct unit of measurement for temperature (as 80 °C"). Use dash for negative temperature instead of hyphen. Use hyphen to denote "to" (as "4–5").

Use "s" as abbreviation for "second" instead of "sec".

In 4.4, do not capitalize "kinetin" and "zeatin". 

For 4.5, 5 min is not long enough period of time for all reaction centers to become open, darkening period needs to be at least 15–20 min depending on the model species.

In 4.15, "analytical replicates" seems to be the right term.

Always use past tense when describing performed actions.

Always use italics for scientific names of plants.

Author Response

Answers to Reviewer 1

 We are very grateful to the comments and suggestions. We carefully addressed each point and changed the manuscript accordingly. This substantially improved our manuscript.

 The obtained results are interesting and the discussion is convincing. However, presentation and scientific style can be significantly improved to increase perceptibility and overall quality of the paper.

 Do not use abbreviations in the title (HIPP, ABA).

We changed the title.

Rephrase as "...in a crosstalk between..."

            We rephrased the sentence.

Abstract

Contains unexplained abbreviations.

            We now use the full name for each abbreviation for the first time that it is mentioned.

Do not divide compound words with a hyphen in the Abstract and further in the whole text. It is "crosstalk", "upregulated", "overexpression", "nonliving" (although better use "physicochemical"), "feedback", "photomorphogenesis", "posttranslational", "downregulated", "multifunctional", "upregulation", "setup". However, it is "ever-changing", "well-known", "trade-off", "well-watered", "drought-induced", "fine-tuning" as used correctly.

            Thanks! We corrected the abovementioned words.

Introduction

The aim needs to be formulated in more detailed manner, including also model species.

We now formulated the aim more precisely, pointing out that we want to analyze function of HIPP protein HvFP1 in crop plants (in contrast to most work on HIPPs done in model plant A. thaliana), especially during development and in response to drought.

"Normal" is not a scientific term, probably "optimal conditions" or "well-watered" are meant.

            We changed the term.

In the last paragraph, no need to describe obtained results in detail, moving even to a discussion.

A brief reference to main results was added in the last paragraph, according to the guidelines for manuscript submission by Plants. We now shortened this paragraph and took out discussion.

Do not introduce abbreviations that are not used further in the text (as NLS).

            We removed the redundant abbreviations.

Results

Do not use "das" as abbreviation for "days after sowing", it is confusing, use "DAS" when necessary.

            We substituted “das” with “DAS”.

Use past tense when describing the results (lines 83-84, lines 87–90, line 94, line 101, line 111, line 122, line 124–126, line 129, line 136–143, lines 151–157, lines 163–165, lines 168–173, lines 223–225).

            We used past tense to present the results.

Do not use references to literature (line 133).

            We removed the literature from results.

Indication of statistically significant differences in Figure 1, Figure 3, Figure 6 seems to be incorrectly used and are confusing, better report results of pairwise post-hoc analysis as different letters, when p < 0.05.

The statistical significance was calculated with the Pair Wise Fixed Reallocation Randomisation Test, which is integrated in version 2 of REST-384 test 2006 (based on Pfaffl et al., 2002; DOI: 10.1093/nar/30.9.e36). This is a widely used software, cited more than 8.000 times, including the works of Wang et al. (2002; DOI: https://doi.org/10.1007/s11033-022-07794-3) and Matos et al (2022; DOI: https://doi.org/10.1016/j.gene.2022.146377), in which the relative gene expression and significant differences are presented in similar manner as in our work. We now further explained in Figure captions the compared samples and made clear which significance is represented by the asterisks.

In Figure 2, Figure 3B, legends on X axis should be "Time after sowing (days)".

            We changed the X axis from “days after sowing (DAS)” to “time after sowing (days)”

These graphs are too small for optimum perception. 

            We improved the size of figures

Section 2.4 contains redundant experimental details belonging to Materials and methods.

            We moved some details to Materials and Methods.

In Discussion, use past tense when describing the results obtained in the present study.

            We now used past tense to discuss our results.

Materials and methods

The type of germination conditions can be indicated as "thermoperiod" (lines 479–480).

            We substituted the word “rhythm” with “thermoperiod”

For a drought stress experiment (and other separate experiments), how many times exactly these experiments were performed (it cannot be "at least three times"). Or does this mean that there were three biological replicates in a single experiment?

Each experiment was definitely performed three times, at independent time points. If necessary (for example if measurement at one specific time point was disrupted or if additional plant material was need), one experiment was performed an additional 4th time. We changed the text to “…experiment was performed three times….”. 

Use "." for decimals instead of ",".

            We substituted “,” with “.”

Use correct unit of measurement for temperature (as 80 °C").

            We corrected the temperature unit

Use dash for negative temperature instead of hyphen. Use hyphen to denote "to" (as "4–5").

            We corrected the dash and hyphens, as suggested

Use "s" as abbreviation for "second" instead of "sec".   

            We corrected the unit

In 4.4, do not capitalize "kinetin" and "zeatin". 

            We changed the capital letters to small letters.

For 4.5, 5 min is not long enough period of time for all reaction centers to become open, darkening period needs to be at least 15–20 min depending on the model species.

We are aware that the recommended time for dark adaptation is 15-20 min. There are also studies, in which 10 min were applied (Schreiber et al., 1995; DOI: 10.1007/978-3-642-79354-7_3; Govindjee & Spilotro, 2002; DOI: 10.1071/PP01136). For our experiments, we employed a MINI-PAM fluorometer with appropriate Dark Leaf Clips for dark adaptation, which weighs approx. 4g. Since we handled very sensitive drought stressed and senescing leaves, we wanted to eliminate the additional mechanical stress and damage on barley primary leaves. For that reason, we applied a minimum of 5min dark adaptation time (as performed also by Hwang et al., 2003; DOI: 10.1007/BF03030441 and Pipitone et al., 2021; DOI: https://doi.org/10.7554/eLife.62709), which still allowed us to get valid measurements for the efficiency of photosystem II. Kobayakawa & Imai (2013; DOI: http://dx.doi.org/10.4236/ajps.2013.49215) noted that after 5-10min of dark adaptation of rice leaves, Fv/Fm ratio remains stable. We checked this also for barley and found that after 5 min values were stable.

In 4.15, "analytical replicates" seems to be the right term.

             We substituted the word “technical” with the word “analytical” in line 588.

Always use past tense when describing performed actions.

             We corrected the tense when needed.

Always use italics for scientific names of plants.

Gene names were indicated with italics in the originally submitted manuscript. Due to unknown reason, species and gene names do not appear in italics in the uploaded manuscript. We will carefully check this in the final version.

Reviewer 2 Report

This quality of the manuscript is not suitable for publication as it stands.

My comments are as follows:

1. The quantitative data in the article is not repetitive well, especially Fig.1C and D miss the control&Mock treatment. 

2. The information on C and D in Fig.2C are missing.

3. The authors should try careful draw the conclusion that over-expression of HvFP1 delays the expression of senescence-associated genes but also drought stress-related genes since the qRT-PCR results of the two hyperexpression lines are frequently inconsistent in Fig.3

4. It is hard to believe that the number of differentially expressed genes detected by RNA-seq is tiny, which is not enough to support their conclusions. The authors should check the quality of the samples for RNA-seq.

Author Response

Answers to Reviewer 2

We are grateful to comments of Reviewer 2. We changed the text accordingly, to eliminate possible misunderstandings.

This quality of the manuscript is not suitable for publication as it stands.

My comments are as follows:

  1. The quantitative data in the article is not repetitive well, especially Fig.1C and D miss the control&Mock treatment.

As described in 4.4, phytohormones ABA, SA and MeJA are dissolved in EtOH solution and then diluted in tap water, in final volume 50ml. For control/mock treatment we used the same final concentration of EtOH in 50ml tap water; without phytohormone. The same applies for cytokinins, but instead of EtOH we used KOH. In Figure 1C and 1D, the relative expression level of HvFP1 in control/mock treatments are not shown separately because they are used as controls (set as 1) and we presented the expression of HvFP1 in response to phytohormones and in comparison, to that in mock controls. We also performed the same analysis, where all mock controls and treatments were compared with untreated leaves, but we didn’t show it in the manuscript because no significant differences were observed. To clarify this, we changed the text in Figure legend.

  1. The information on C and D in Fig.2C are missing.

We specified the meaning of the letters in figure caption.

  1. The authors should try careful draw the conclusion that over-expression of HvFP1 delays the expression of senescence-associated genes but also drought stress-related genes since the qRT-PCR results of the two hyperexpression lines are frequently inconsistent in Fig.3

In our experimental design, we followed the progress of drought-induced and developmental leaf senescence at multiple time points and stages. Each time point/stage is characterized by specific physiological and transcriptomic changes (Guo & Gan, 2012; DOI: 10.1111/j.1365-3040.2011.02442.x; Bhargava & Sawant, 2013; DOI: doi:10.1111/pbr.12004; Guo et al., 2021; DOI: https://doi.org/10.1186/s43897-021-00006-9). In both experiments (for each treatment, three INDEPENDENT experiments were performed), we see a clear effect in both OE lines. This is that during developmental senescence, starting when chlorophyll content is clearly decreasing, i.e., being lower than 95% (= 90, 80, 75 or 50%), expression levels are ALWAYS lower in both OE lines compared to WT. Same is shown for drought treatment: expression of the three stress-related genes HvS40, HvNCED and HvDhn1, when plants are exposed to drought (starting 8 days after stop of irrigation when soil water content dropped to about 50% of control) is clearly lower than in WT, for both OE lines. Some inconsistencies in expression levels, in our opinion, can be justified due to natural variations when comparing individual plants in independent experiments during different times under senescence and stress conditions.

Therefore, we think that our conclusion that over-expression of HvFP1 delays the expression of senescence-associated genes but also drought stress-related genes, is justified.

  1. It is hard to believe that the number of differentially expressed genes detected by RNA-seq is tiny, which is not enough to support their conclusions. The authors should check the quality of the samples for RNA-seq.

The quality of samples was checked prior the RNASeq with a nanospectrophotometer and a bioanalyzer instrument, which provided the concentration of RNA, the integrity of the subunits of ribosomal RNA, the ratio of 25S and 18S rRNA and the RNA integrity number (RIN) for each sample. Only good quality (RIN value > 5) and high amount of RNA samples (>40 ng/µl) were sent for sequencing. The company repeated the quality control of the samples and every step forward (library preparation and sequencing) was performed together with the corresponding quality control, in order to ensure the optimal outcome.

The experimental approach includes samples of three independent biological replicates. Only the differentially expressed genes in all replicates, with a log2foldchange > 2 and adjusted p value > 0.05 are used for further analysis. These factors justify the relatively small number of differentially expressed genes. One has to take into consideration that the present manuscript deals with the differentially expressed genes in HvFP1 OE line in comparison to WT samples. The analysis of WT or OE control samples with those in senescence state results in a much higher number of genes (also known as senescence associated genes), but they are not subject of the present work. Still, taking into consideration the incomplete genome annotation and the limited research on barley, total 123 differentially regulated genes in one mutant line in comparison to WT are novel for this model crop plant and adequate for making conclusions about a possible function of HvFP1.

Reviewer 3 Report

In this manuscript, the authors provided novel data to illustrate the possible function of HvFP1 in barley. The data are sufficient and the story is interesting. It is of good help for understanding the role of HIPP protein members, which was rarely investigated.

Nevertheless, there are some points need to be addressed or discussed:

1. How can HvFP1 regulate functional gene expression? The authors provided data about gene expression, which may be result from overexpression of HvFP1. They also discussed the possibility of interaction and zinc transferring from HvFP1 to some zinc-binding proteins. To build a full story, how can senescence and phytohormone regulate expression of HvFP1 has been illustrated, while how can HvFP1 promote functional gene expression is still absent.

2.The discussion is not concise enough. Some repeated sentences describing results which had been showed in the Result section need to be deleted.

Author Response

Answers to Reviewer 3

We are thankful to the comments and suggestions of Reviewer 3. They help a lot to improve the quality of the manuscript.

In this manuscript, the authors provided novel data to illustrate the possible function of HvFP1 in barley. The data are sufficient and the story is interesting. It is of good help for understanding the role of HIPP protein members, which was rarely investigated.

Nevertheless, there are some points need to be addressed or discussed:

  1. How can HvFP1 regulate functional gene expression? The authors provided data about gene expression, which may be result from overexpression of HvFP1. They also discussed the possibility of interaction and zinc transferring from HvFP1 to some zinc-binding proteins. To build a full story, how can senescence and phytohormone regulate expression of HvFP1 has been illustrated, while how can HvFP1 promote functional gene expression is still absent.

The Reviewer is right that we do not present a final model of mechanism how HvFP1 promotes functional gene expression. Nevertheless, this is the first study, in which it is shown that a HIPP protein in the model crop plant Hordeum vulgare affects major developmental and stress-related pathways via reprogramming of gene expression. We present data showing that OE of HvFP1 results in induction of well-known transcription factors/activators (like zinc binding CCCH and Far1-related sequences). Then, these genes regulate downstream major plant processes (Wang & Wang, 2015; DOI: http://dx.doi.org/10.1016/j.tplants.2015.04.003; Ma & Li, 2018; DOI: 10.3389/fpls.2018.00692; Ai et al., 2022; DOI: https://doi.org/10.1186/s12870-022-03500-4).

As discussed in the manuscript, homologous genes in Arabidopsis and quinoa were found to interact with zinc finger transcription factors for the downstream regulation of drought response genes (Barth et al., 2009; DOI: 10.1007/s11103-008-9419-0; Sun et al., 2022; DOI: https://doi.org/10.1016/j.plantsci.2022.111406), while another Arabidopsis HIPP is known to bind zinc (Zschiesche et al., 2015; DOI: 10.1111/nph.13419). In our work, there are severe indications that HvFP1 follows the same mode by inducing, probably by supplying zinc, some zinc binding transcription factors/activators for downstream regulation of genes. However, this is discussed as a hypothesis and not yet proven. In addition, this is the first study, in which an RNA seq is performed in barley HIPP mutants, giving us more insight into possible interaction partners and downstream regulated factors. Of course, future experiments are needed to unravel the molecular mechanism of HvFP1 function. Nevertheless, we think that our results are important to better understand the role of HIPP proteins. 

2.The discussion is not concise enough. Some repeated sentences describing results which had been showed in the Result section need to be deleted.

Thanks for this comment. We proofread and removed the redundant sentences, which clearly improves quality of the discussion.

Reviewer 4 Report

The manuscript Plants-1953690 deals with very interesting and important topic about HvFP1 gene study and the interaction with plant hormones in barley plants in response to drought. The manuscript is very well written with valuable results and it can be accepted in general after minor-moderate revision. I would like to see authors’ responses to the major points indicated below

Major comments/suggestions:

(1) L538-548. My main concern is about the description of the used qPCR method. Authors indicated “four dilution series for each sample, corresponding to 1:4, 1:16, 1:64 or 1:256”. This is very strange statement because these are neither biological replicates (because are made from the same sample) nor technical replicates (because these are diluted samples). What authors want to say using these unnecessarily dilutions of all samples? This is unclear. The only my possible reasonable explanation is that authors want to mention a serious of dilution of three used reference genes (HvPP2A, HvActin and HvGCN5) for the calibration standard curve (or line) based on a serious of dilution of PCR products of known reference genes. Authors should find (if not yet) and refer to “Example of the standard curve method (Singleplex)…” (pp. 36-38) from the ABI ‘Guide for relative quantitation of gene expression using real-time quantitative PCR’ from their web-site (http://www3.appliedbiosystems.com/cms/groups/mcb_support/documents/generaldocuments/cms_042380.pdf). Another sources are also acceptable where this method with a serious of dilution has been described. Many authors use serious of such dilution for reference gene (or genes) and only once. After that all studied target genes as well as reference genes are compared with the standard calibration curve (or line). Usually, people calculate and prepare 1 femtomole (10^-15 mole) dilution of PCR products of Reference gene(s) with following consequent dilutions for 100; 10,000; and 1,000,000 folds, and added these diluted PCR products instead cDNA template for qPCR analysis for calibration purposes. In this case, there is no necessity to make a serious of four dilutions for each studied sample because KAPA reagents and qPCR consumables are very expensive in general. Please clarify and insert your explanation in the text of the revised and corrected section.

(2) L548 and Supplementary Table S3. The information about Reference genes is unacceptable. Firstly, I strongly recommend authors to add citations for the publications where these Reference genes were studied and verified as suitable for qPCR analysis. If no published information is available for any of the used Reference genes, please provide your statement why authors believe that these three Reference genes are the best choice for their qPCR experiments. Additionally, there are many (or at least several) HvActin genes in barley. Therefore, only sequences of the used primers are not sufficient. Authors MUST provide full information about all used genes (Reference and target) at least in the Supplementary Table S3 including the follows: Abbreviated and full name of gene, ID from public database (preferably from NCBI), and amplicon size for each 17 studied genes. Additionally, please delete the second (doubled) Table S3 due to unnecessary repeats. Finally, it would be also perfect to add one more Supplementary material with sequences of all 17 genes used for qPCR, where primer sequences are indicated by colour or font. However, this last point is not compulsory if authors do not like it.

Minor notes/corrections:

(3) There is massive negligence of authors in entire manuscript, where ALL names of genes MUST be written in Italics. I started spotting it from L81: “Expression of HvFP1 was analysed…” but names of genes are present not in Italics everywhere in the text! This conflict is especially clear for example in the legend of Figure 1 (L95-96), where the name of gene is written in Italics: “…transcript level of HvFP1 in samples…” but not in the text. The name ‘HvFP1’ in normal (non-Italics) case is only used for polypeptides and proteins, which authors correctly indicated and used in Supplementary Figure S1-A, right part with Western blot. In all other cases, Authors must correct all gene names in Italics unless they mention polypeptides.

(4) L83-83 and further in text and Figures. Authors used the abbreviation ‘das’ for ‘Days after sowing’. However, in current form, ’das’ looks as misspelling words. All other people use ‘DAS’ in upper case indicating for the abbreviation. This is not a scientific point but it would be much easier to understand ‘13th DAS’ rather than ‘13th das’. Please correct.

(5) L95-96. Legend of Figure 1. Authors mentioned “control treatment with pure ethanol” and with “KOH”. However, in current form, these statements are shocking and mislead readers that authors cut leaves and put in ‘pure ethanol’ or in ‘KOH’. I know and authors perfectly described their treatments in M&M that ABA, SA and MeJA were just dissolved in ethanol and after that diluted until required concentrations, and similar is about other hormones dissolved in KOH. However, in the current form, the Legend of Figure 1 is horrible. I recommend just delete your reference to ethanol and KOH because it is clearly described in M&M. Alternatively, if authors wish, please write as follows: ‘…in comparison to control treatment without hormones…’. Both variants are fine and understandable for readers.

(6) L95-96. Legend of Figure 1A needs improvement. X-axis in the Figure represents ‘Chlorophyll content’ ranged from 100% to <50% leaves but in the corresponding Legend is written as follows: “…relative transcript level …during different stages of developmental leaf senescence…”. Authors must modify either Figure legend or X-axis in the Figure making complete match between each other. Additional explanation must be provided in the text.

(7) All Figures and statistical treatment. Authors presented their error bars as SD (Standard deviation) but not SE (Standard error). This is OK and not a mistake; and this is also in the authors’ preference, how they wish to present their statistical treatments. However, SD bars so big in all Figures, especially in Figure S1-A and S2 with big differences between samples, which makes the Figure analyses irritating for readers. I would suggest authors to re-calculate SD and change it for SE, which will be about three-fold smaller and easier to see for readers. However, this is only if authors wish to do so.

(8) L121, L147-148, L158-159, L162, L170, Please remove unclear term ‘marker genes’ and replace it for more suitable ‘senescence-associated genes’ or ‘drought stress-related genes’ as authors correctly used in L156 and L166, respectively.

(9) L207. Similar to point 4 above, Please use better spelling with Upper case ‘IDs’ (Identifications) as follows ‘The gene IDs and…’ but not “The gene ids and…”.

(10) L209, L212, L420, L437 and maybe in other places. All botanical names of plant species in Latin language must be in Italics. The exception might be made only for ‘Arabidopsis’ (L416) if authors want to say it in English.

(11) L479-480. The following phrase is unclear: “…in a 16 h/8 h rhythm for 48 h in dark”. Why 48 but not 24 hours are indicated? Why 48 h in dark? The term ‘rhythm’ should be replaced as not suitable in this context.

(12) L589-590. Please clarify and indicate in this fragment that ‘qPCR analysis was performed with three biological replicates for each independent experiment’ (similar to those indicated for RNA-seq in L594). Four technical replicates are mentioned perfectly.

Author Response

Answers to Reviewer 4

We are very thankful for the helpful comments and suggestions of Reviewer 4. They help to improve the quality of our manuscript.

The manuscript Plants-1953690 deals with very interesting and important topic about HvFP1 gene study and the interaction with plant hormones in barley plants in response to drought. The manuscript is very well written with valuable results and it can be accepted in general after minor-moderate revision. I would like to see authors’ responses to the major points indicated below

Major comments/suggestions:

(1) L538-548. My main concern is about the description of the used qPCR method. Authors indicated “four dilution series for each sample, corresponding to 1:4, 1:16, 1:64 or 1:256”. This is very strange statement because these are neither biological replicates (because are made from the same sample) nor technical replicates (because these are diluted samples).

We replaced the term “technical” with “analytical” replicates in 4.15.

What authors want to say using these unnecessarily dilutions of all samples? This is unclear. The only my possible reasonable explanation is that authors want to mention a serious of dilution of three used reference genes (HvPP2AHvActin and HvGCN5) for the calibration standard curve (or line) based on a serious of dilution of PCR products of known reference genes. Authors should find (if not yet) and refer to “Example of the standard curve method (Singleplex)…” (pp. 36-38) from the ABI ‘Guide for relative quantitation of gene expression using real-time quantitative PCR’ from their web-site (http://www.appliedbiosystems.com/cms/groups/mcb_support/documents/generaldocuments/cms_042380.pdf ). Another sources are also acceptable where this method with a serious of dilution has been described. Many authors use serious of such dilution for reference gene (or genes) and only once. After that all studied target genes as well as reference genes are compared with the standard calibration curve (or line). Usually, people calculate and prepare 1 femtomole (10^-15 mole) dilution of PCR products of Reference gene(s) with following consequent dilutions for 100; 10,000; and 1,000,000 folds, and added these diluted PCR products instead cDNA template for qPCR analysis for calibration purposes. In this case, there is no necessity to make a serious of four dilutions for each studied sample because KAPA reagents and qPCR consumables are very expensive in general. Please clarify and insert your explanation in the text of the revised and corrected section.

For the qRT-PCR analysis, we followed the method described by Pfaffl et al. (2002; DOI: 10.1093/nar/30.9.e36), which has been cited in more than 8.000 scientific articles. Specifically, we used the updated version 2 REST-384 test 2006. This software is capable of calculating the relative expression level of multiple reference and target genes in two groups of samples, with up to 20 data points for each group. At the same time, it runs a Pair Wise Fixed Reallocation Randomisation Test, which gives the significant differences between the groups for each tested gene. One necessary parameter for this test is the efficiency of primer pairs of reference and target genes. For that reason, we employed the mentioned four series of dilutions and used the outcome Crossing Point (CP) values from a CFX Connect Real-Time PCR Detection System to calculate the efficiency (E) of the primers as E=10(-1/slope), where slope is calculated by the CP and the log(Starting Quantity). In optimal conditions, E=2 and the CP values among dilution series differ by 2. So, the dilution series we used is just an additional step to ensure that efficiency of primer pairs is good (this test is quite often not performed by researchers since for many (but not all) primer pairs employed, efficiency is often close to 2). We added this information to 4.9. The advantage of the used software is that we were able to normalize the relative expression level of multiple genes with three reference genes and perform a statistical analysis with a randomization test, taking into consideration all possible combinations between control and treatment values.

(2) L548 and Supplementary Table S3. The information about Reference genes is unacceptable. Firstly, I strongly recommend authors to add citations for the publications where these Reference genes were studied and verified as suitable for qPCR analysis. If no published information is available for any of the used Reference genes, please provide your statement why authors believe that these three Reference genes are the best choice for their qPCR experiments.

Thanks for this helpful comment. Among the three reference genes, which were used in the present study, Protein Phosphatase 2A (PP2A) and actin have been evaluated and approved for qRT-PCR normalization, even under various developmental and abiotic stress conditions (Chen et al., 2015; DOI: 10.1007/s00299-015-1830-9; Sudhakar Reddy et al., 2016; DOI: 10.3389/fpls.2016.00529; Gines et al., 2018; DOI: 10.2135/cropsci2017.07.0443). We added the references to the text in 4.9. The third gene, named General control nonderepressible 5 (GCN5) is a histone deacetylase with no reported use as reference gene, so far. However, the expression of this gene in barley primary leaves, in all our samples and treatments, was stable, thus was further used as reference gene for normalization of qRT-PCR. We added information in the text (in chapter 4.9.) that expression of this gene was stable under all conditions used in this manuscript.

Additionally, there are many (or at least several) HvActin genes in barley. Therefore, only sequences of the used primers are not sufficient. Authors MUST provide full information about all used genes (Reference and target) at least in the Supplementary Table S3 including the follows: Abbreviated and full name of gene, ID from public database (preferably from NCBI), and amplicon size for each 17 studied genes.

The used HvActin reference gene corresponds to HvActin7 (gene ID: HORVU1Hr1G002840). We also added all the requested information in Supplemental Table S3.

Additionally, please delete the second (doubled) Table S3 due to unnecessary repeats.

            We deleted the repeated table

Finally, it would be also perfect to add one more Supplementary material with sequences of all 17 genes used for qPCR, where primer sequences are indicated by colour or font. However, this last point is not compulsory if authors do not like it.

We now provided gene names and gene IDs, which can be used to retrieve the gene sequences from NCBI, IPK Barlex or Ensembl Plants websites.

Minor notes/corrections:

(3) There is massive negligence of authors in entire manuscript, where ALL names of genes MUST be written in Italics. I started spotting it from L81: “Expression of HvFP1 was analysed…” but names of genes are present not in Italics everywhere in the text! This conflict is especially clear for example in the legend of Figure 1 (L95-96), where the name of gene is written in Italics: “…transcript level of HvFP1 in samples…” but not in the text. The name ‘HvFP1’ in normal (non-Italics) case is only used for polypeptides and proteins, which authors correctly indicated and used in Supplementary Figure S1-A, right part with Western blot. In all other cases, Authors must correct all gene names in Italics unless they mention polypeptides.

We were very careful and used always italics for genes and species names in the originally submitted manuscript. However, during uploading of the manuscript to the Journals website, the formatting unintentionally changed. We also noticed the mistakes in font in the uploaded version of the manuscript and will carefully check this for the final version.

(4) L83-83 and further in text and Figures. Authors used the abbreviation ‘das’ for ‘Days after sowing’. However, in current form, ’das’ looks as misspelling words. All other people use ‘DAS’ in upper case indicating for the abbreviation. This is not a scientific point but it would be much easier to understand ‘13th DAS’ rather than ‘13th das’. Please correct.

            We replaced “das” with “DAS”

(5) L95-96. Legend of Figure 1. Authors mentioned “control treatment with pure ethanol” and with “KOH”. However, in current form, these statements are shocking and mislead readers that authors cut leaves and put in ‘pure ethanol’ or in ‘KOH’. I know and authors perfectly described their treatments in M&M that ABA, SA and MeJA were just dissolved in ethanol and after that diluted until required concentrations, and similar is about other hormones dissolved in KOH. However, in the current form, the Legend of Figure 1 is horrible. I recommend just delete your reference to ethanol and KOH because it is clearly described in M&M. Alternatively, if authors wish, please write as follows: ‘…in comparison to control treatment without hormones…’. Both variants are fine and understandable for readers.

Thanks. To avoid misunderstanding, we rephrased the description of control treatments in the caption of Figure 1

(6) L95-96. Legend of Figure 1A needs improvement. X-axis in the Figure represents ‘Chlorophyll content’ ranged from 100% to <50% leaves but in the corresponding Legend is written as follows: “…relative transcript level …during different stages of developmental leaf senescence…”. Authors must modify either Figure legend or X-axis in the Figure making complete match between each other. Additional explanation must be provided in the text.

            We clarified this point in the caption of Figure 1

(7) All Figures and statistical treatment. Authors presented their error bars as SD (Standard deviation) but not SE (Standard error). This is OK and not a mistake; and this is also in the authors’ preference, how they wish to present their statistical treatments. However, SD bars so big in all Figures, especially in Figure S1-A and S2 with big differences between samples, which makes the Figure analyses irritating for readers. I would suggest authors to re-calculate SD and change it for SE, which will be about three-fold smaller and easier to see for readers. However, this is only if authors wish to do so.

            Thanks for this comment. We now calculated the SE and replaced SD in all graphs

(8) L121, L147-148, L158-159, L162, L170, Please remove unclear term ‘marker genes’ and replace it for more suitable ‘senescence-associated genes’ or ‘drought stress-related genes’ as authors correctly used in L156 and L166, respectively.

We replaced the term “marker gene” with “senescence-associated genes” or “drought stress-related genes”

(9) L207. Similar to point 4 above, Please use better spelling with Upper case ‘IDs’ (Identifications) as follows ‘The gene IDs and…’ but not “The gene ids and…”.

            We replaced “ids” with “IDs”

(10) L209, L212, L420, L437 and maybe in other places. All botanical names of plant species in Latin language must be in Italics. The exception might be made only for ‘Arabidopsis’ (L416) if authors want to say it in English.

Gene names were indicated with italics in the originally submitted manuscript. Due to unknown reason, species and gene names do not appear in italics in the uploaded manuscript. We will carefully check this in the final version.

(11) L479-480. The following phrase is unclear: “…in a 16 h/8 h rhythm for 48 h in dark”. Why 48 but not 24 hours are indicated? Why 48 h in dark? The term ‘rhythm’ should be replaced as not suitable in this context.

            We replaced the word “rhythm” with the word “thermoperiod”.

(12) L589-590. Please clarify and indicate in this fragment that ‘qPCR analysis was performed with three biological replicates for each independent experiment’ (similar to those indicated for RNA-seq in L594). Four technical replicates are mentioned perfectly.

            We specified the performance of three independent biological replicates

Round 2

Reviewer 2 Report

The second revision has improved the manuscript considerably. 

Reviewer 4 Report

Authors made their great job and addressed all questions propercly. I have no any further comment.